# Has the Development of the Non-Timber Forest Products Industry Achieved Poverty Alleviation? Evidence from Lower-Income Forest Areas in Yunnan Province

**Yaquan Dou** [1] , **Jian Wu** [1]**, Ya Li** [2]**, Xingliang Chen** [3] **and Xiaodi Zhao** [1,*]

[1] Research Institute of Forestry Policy and Information, Chinese Academy of Forestry, Beijing 100091, China; douyq@caf.ac.cn (Y.D.)
[2] School of Economics and Management, Southwest Forestry University, Kunming 650224, China; liy@swfu.edu.cn
[3] Non-Timber Forest-Based Economy Branch, Chinese Society of Forestry, Beijing 100091, China; chen62889299@126.com
\* Correspondence: zhaoxiaodi@caf.ac.cn; Tel.: +86-136-7105-3260

**Abstract:** Considering the notion that "lucid waters and lush mountains are invaluable assets", the effective exploitation of the economic value of forest resources is an important research topic, especially in forest-rich areas. The development of the non-timber forest products (NTFPs) industry has promoted both ecological and economic benefits and has effectively improved farmers' incomes while protecting forest resources. In order to evaluate the effects of the NTFPs industry on sustaining farmers' livelihoods and protecting ecological environments, we constructed a performance evaluation index system to determine the poverty alleviation performance of the NTFPs industry in Yunnan Province using the analytic hierarchy process (AHP), which covered three aspects: the achievement of poverty alleviation, the sustainability of poverty alleviation and satisfaction with poverty alleviation. Then, we selected Sanhe Village in Nujiang Prefecture, Yunnan Province, as an example to verify and rationalize the evaluation index system and comprehensively evaluate the poverty alleviation performance of the NTFPs industry. Based on data from questionnaires and field interviews, we found the following: (1) the overall poverty alleviation performance of the NTFPs industry in Sanhe Village was 79.33, which indicated that the effect was good; (2) the scores for the achievement of poverty alleviation, the sustainability of poverty alleviation and satisfaction with poverty alleviation were 50.56, 18.57 and 10.2, respectively; (3) there were some problems with the poverty alleviation performance of the NTFPs industry, such as limited capital investments, the weak roles of cooperatives and enterprises, the low enthusiasm of lower-income households and incomplete poverty alleviation projects. Finally, we devised some suggestions that could improve the poverty alleviation performance of the NTFPs industry. This paper presents the performance evaluation index system for the poverty alleviation performance of the NTFPs industry, which could provide a reference for evaluating the developmental effects of the NTFPs industry in other lower-income forest areas. Through our empirical analysis of the development effects of the NTFPs industry on farmers' livelihoods and ecological environments in Sanhe Village, we found that the development of the NTFPs industry significantly improved the farmers' livelihoods and ecological environments.

**Keywords:** sustainable forest management; NTFPs industry; protection and development; AHP; comprehensive evaluation method

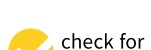



## 1. Introduction

With the increasing seriousness of climate change and biodiversity loss, the roles of forest resources in carbon fixation, oxygen release, water conservation and biodiversity maintenance are becoming more obvious [1]. Forest resources are basic resources for human survival and have various functions, such as water conservation, carbon sequestration



and oxygen release, as well as biodiversity protection [2,3]. At the same time, forest resources continue to provide humans with rich material products, such as fuel, food and medicine [4,5]. Therefore, the protection and utilization of forest resources have become major concerns for policymakers and managers [6]. Recently, China has successively implemented a number of key ecological projects, such as Returning Farmland to Forests and Protecting Natural Forests, to effectively protect forest resources [7]. However, the implementation of these projects has restricted the economic development of forest areas [8]. In this context, how to make full use of rich forest resources to realize regional economic development and how to transform forest resources into economic advantages have become the focus of research in lower-income forest areas [9].

Yunnan Province, located on the southwest border of China, is an important node province in the implementation of the Belt and Road Initiative, which uses the Silk Road Economic Belt and 21st Century Maritime Silk Road to promote regional development, and was a pioneer in the construction of ecological civilization in China. By the end of 2021, the forest areas in Yunnan Province covered 25 million ha, making Yunnan the second most forested province in China. Obviously, forest resources are among the most important sources of production and living materials for farmers in Yunnan Province and are also important for the economic development of forest areas [10,11]. Therefore, it is vital to make full use of forest resources to develop forestry industries in order to achieve high-quality economic development and ecological civilization in Yunnan Province. It is common for Chinese forest farmers to hold forest rights certificates but still have lower income [12]. Especially in Yunnan Province, collective forests have been classified into natural forest reserves and nature reserves, resulting in the heavy restriction of farming activities [13]. In order to effectively alleviate the contradiction between ecological protection and the sustainability of farmers' livelihoods, forest farmers have made full use of forest resources and have vigorously developed characteristic green industries in Yunnan Province [14]. In recent years, forest farmers in Yunnan Province have positively participated in the non-timber forest products (NTFPs) industry, mainly in the *Amomum tsao-ko* industry, the walnut industry and ecotourism, which has improved farmers' livelihoods while protecting forest resources [15].

The role of NTFPs in sustaining forest-based livelihoods and improving farmers' ability to cope with climate change has been recognized [16–19]. Heubach et al. (2012) found that income from NTFPs accounted for 39% of total household income and had a strong equalizing effect on total household income in northern Benin, according to survey data from 230 households in two villages [20]. Similarly, Mukul et al. (2016) found that 27% of households in a protected area in Bangladesh received at least some income from the collection, processing and sale of NTFPs and that NTFPs contributed to primary, supplementary and emergency sources of household income. NTFPs also constituted an estimated 19% of household net annual income and were the primary occupation for about 18% of households [21]. At the same time, NTFPs management has led to various benefits for community livelihoods and forest sustainability in Indonesia and the Northern Zagros in Iran [22,23]. In general, scholars have carried out a great deal of research on the benefits of the NTFPs industry in terms of ensuring food security, improving local livelihoods and reducing poverty rates [24–26].

With the transformation and upgrade of forest-related industries and the need for high-quality development, some regions in China are continuing to explore the development of the NTFPs industry. It has been proven that the NTFPs industry can improve farmers' incomes, promote regional economic development and stimulate other economic functions [27]. Moreover, the ecological functions of maintaining the stability and diversity of forest ecosystems have also proven to be effective [28]. At the same time, scholars have carried out a series of studies on the advantages and disadvantages of the NTFPs industry [29], its benefits [30] and its impacts on ecological environments [31]. With the deepening of the reform of collective forest rights systems, more and more studies have focused on the NTFPs industry; however, these studies have mainly focused on summarizing

development models and introducing typical cases in various regions. There have been relatively few studies on the combination of the industrial development of NTFPs and poverty alleviation. Over the years, some regions have explored effective ways of promoting the industrialization of NTFPs. By the end of 2021, forest areas in Yunnan Province covered 20.2 million ha, with a forest coverage rate of 65.4%, which was a significant increase from previous years. As a major forestry province in China, Yunnan has explored effective ways of promoting the development of the NTFPs industry for many years, but there are still some questions that remain unanswered: How effective has the industry been over the years? Are farmers really benefiting? Is the NTFPs industry sustainable? These questions need in-depth study and analysis. Consequently, we constructed a performance evaluation system for the poverty alleviation performance of the NTFPs industry, which could provide a reference for other regions in Yunnan Province. Additionally, we selected Sanhe Village as a case study for our empirical analysis to evaluate the impact of the NTFPs industry on farmers' livelihoods and ecological environments through a literature review, data collection and field research. By analyzing and evaluating the effects of the poverty alleviation performance of the NTFPs industry in Sanhe Village, we also identified shortcomings and problems in the process of poverty alleviation. Finally, we devised some suggestions that could improve the poverty alleviation performance of the NTFPs industry so as to further improve forestry productivity and household livelihoods in Sanhe Village. Therefore, this paper could not only help to transform and upgrade the NTFPs industry but could also have important practical significance for helping to alleviate poverty in forest areas.

## 2. Materials and Methods

### 2.1. Study Area

Sanhe Village has an area of 6700 ha and a forest coverage rate of 92% and is located in Nujiang Lisu Autonomous Prefecture, Yunnan Province, China. It has jurisdiction over 8 natural villages and 12 villager groups, including more than 1300 people and more than 400 households of Han, Lisu, Nu and Jingpo ethnicity, among others [15,32]. The average altitude of Sanhe Village is 1453 m, the average annual temperature is 16.7 °C and the annual precipitation is 1342 mm, which makes the area suitable for all kinds of crops, trees, traditional Chinese medicines and other NTFPs [33]. In Sanhe Village, the state-owned forest area covers 737.6 ha, the collective commercial forest area covers 2956.2 ha and the per capita forest area covers 2.7 ha. It is an area with very low income in the Gaoligong Mountains, which is itself an area with extremely low income in China. In 2015, there were 369 poverty-stricken people in Sanhe Village and the incidence of poverty was 28% [15,34]. According to our survey, the main causes of poverty were the natural conditions in the village. The area has a harsh climate and often suffers from natural disasters, such as debris flows and floods. However, the risk resilience of farmers is low and natural disasters can cause huge losses in the productivity and livelihoods of farmers. Therefore, due to the climatic and natural conditions in the village, the amount of poverty alleviation resources available to farmers is limited. In this context, rich forest resources have become important resources for farmers in Sanhe Village and also provide an important material basis for reducing poverty in the area.

In recent years, Sanhe Village has vigorously implemented various ecological projects, such as the conversion of farmland back into forest land and natural forest protection. Therefore, forest resources have been effectively protected, which has provided an effective material basis and resources for the development of the NTFPs industry. As of the end of 2021, the government has invested significantly in the development of the NTFPs industry, including a total investment of CNY 2.42 million for under-forest planting and raising. At the same time, farmers have actively participated in the development of the NTFPs industry. As a consequence, a complex management mode of agroforestry has been formed in Sanhe Village. As shown in Table 1, it is obvious that the NTFPs industry has brought certain economic benefits to local farmers and has reduced the poverty rate in Sanhe Village. In particular, there are 866.67 ha of *Amomum tsao-ko* in Sanhe Village, with an annual output

of more than 1.5 million kg, which generates CNY 380,000 of income for the village every year. Based on the above analysis, the poverty problem is serious and available resources are limited in Sanhe Village. However, rich forest resources provide a foundation for the development of the NTFPs industry. At the same time, the government and farmers have high enthusiasm for industrial development and the poverty alleviation effect of NTFPs is obvious. In order to solve the scientific issues in our research question, we selected Sanhe Village as a case study (Figure 1).

**Table 1.** The planting areas of and income from the NTFPs industry in Sanhe Village.

| Number | Type of NTFPs | Planting Area (ha) | Yield (kg/ha) | Income (CNY/ha) | Average Annual Income (CNY Million) |
|---|---|---|---|---|---|
| 1 | *Konjak* | 250.00 | $3.5 \times 10^4$ | $5.0 \times 10^3$ | $1.2 \times 10^6$ |
| 2 | *Mangnolia officinalis* | 13.33 | $3.4 \times 10^3$ | $6.0 \times 10^4$ | $8.0 \times 10^5$ |
| 3 | *Aralia chinensis* | 7.33 | $4.2 \times 10^3$ | $7.5 \times 10^4$ | $5.5 \times 10^5$ |
| 4 | *Paris polyphylla* | 3.33 | $6.8 \times 10^3$ | $1.5 \times 10^6$ | $5.0 \times 10^6$ |
| 5 | *Aucklandia costus* Falc. | 6.67 | $2.8 \times 10^4$ | $4.5 \times 10^4$ | $3.0 \times 10^5$ |
| 6 | *Phellodendron chinense* Schneid. | 33.33 | $4.1 \times 10^4$ | $9.0 \times 10^4$ | $3.0 \times 10^6$ |
| 7 | *Citrus × limon* | 1.00 | $4.5 \times 10^4$ | $9.0 \times 10^4$ | $9.0 \times 10^4$ |
| 8 | Tea | 13.33 | $3.8 \times 10^3$ | $3.0 \times 10^4$ | $4.0 \times 10^5$ |
| 9 | Ecological vegetables | 10.00 | - | $4.9 \times 10^4$ | $4.9 \times 10^5$ |
| 10 | *Amomum tsao-ko* | 13.33 | $7.5 \times 10^3$ | $1.5 \times 10^4$ | $2.0 \times 10^5$ |
| 11 | Walnut | 26.67 | $1.5 \times 10^3$ | $1.5 \times 10^4$ | $4.0 \times 10^5$ |

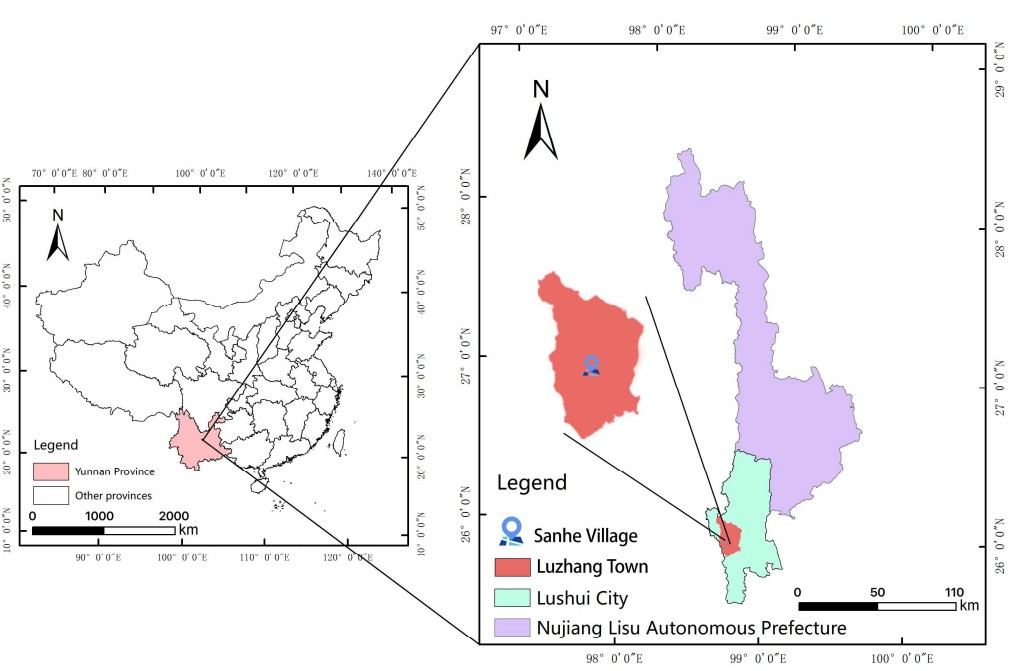

**Figure 1.** The location of Sanhe Village.

## 2.2. Data Sources

We conducted a face-to-face survey in Sanhe Village from April to August 2022. Field observations, questionnaires and in-depth interviews were used in this survey. Firstly, in order to understand the development situation of the NTFPs industry and its poverty alleviation performance, we held in-depth interviews with the relevant heads of the Nujiang Forestry and Grassland Bureau and Poverty Alleviation Office (Appendices A.1.1 and A.1.2 in Appendix A). Secondly, according to the development situation of the NTFPs industry in Nujiang Prefecture, we selected Sanhe Village as the study area to conduct questionnaires. Thirdly, in order to further understand the effects of the NTFPs industry on

farmers' livelihoods, we selected relevant leaders in Sanhe Village for in-depth interviews (Appendix A.1.3 in Appendix A). At the same time, we formed an in-depth knowledge base of the development and poverty alleviation performance of the NTFPs industry through field observations. Fourthly, in order to better analyze the effects of the development of the NTFPs industry on poverty alleviation, we used the judgment sampling method to select lower-income households to complete questionnaires, with the approval and organization of the local government. There are 402 households in Sanhe Village, but the number of lower-income households participating in the NTFPs industry is limited and the households are relatively scattered. Therefore, we finally selected 150 lower-income households for the questionnaire and recovered 139 valid questionnaires, with an effective rate of 92.7%. Additionally, in order to ensure the accuracy and validity of the questionnaire results, we selected one of family members who was familiar with their family's situation and the development of the NTFPs industry. The questionnaire mainly included questions about basic information, financial situation, forest land management, organizational forms and skills training (Table 2).

**Table 2.** The specific questions on the questionnaire.

| Variables | Scope of Variables |
| --- | --- |
| Basic information | Gender, age, ethnicity, education level, etc. |
| Financial situation | Income and expenditure, poverty alleviation policies and measures, etc. |
| Forest land management | Forest land area, non-timber forest products management, etc. |
| Organizational forms | Cooperative organizations, enterprises, etc. |
| Skills training | Willingness to train, content, effects, etc. |

### 2.3. Analytic Hierarchy Process

The analytic hierarchy process (AHP) decomposes decision-making problems into different hierarchical structures, according to the overall objective, the sub-objectives at each level and the evaluation criteria. Then, the eigenvectors of the judgment matrix are solved to obtain the priority weight of each element at each level in relation to an element at the next level. Finally, the weighted sum method is used to merge the final weights of all alternatives to the total objective [35–37]. In this study, we used the AHP to construct a performance evaluation index system for the poverty alleviation performance of the NTFPs industry in Yunnan Province (Figure 2).

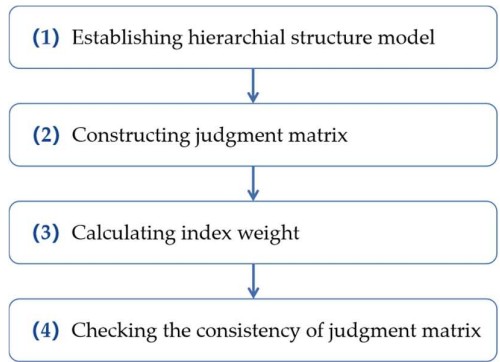

**Figure 2.** The analytical steps of the AHP.

### 2.3.1. Establishing the Hierarchical Structure Model

By using the AHP, we distributed questionnaires to 30 relevant experts from 8 scientific research institutions, including the Chinese Academy of Forestry (CAF), Beijing Forestry University (BFU), Nanjing Forestry University (NFU), Zhejiang Agriculture and Forestry University (ZAFU), Northeast Forestry University (NEFU), Southwest Forestry University (SWFU), Shenyang Agricultural University (SYAU), Fujian Agriculture and Forestry University (FAFU), as well as competent authorities from poverty alleviation departments

in Lushui City and Sanhe Village. Finally, we received a total of 22 valid questionnaires. Then, based on a literature review, we determined the indicators of the index system using answers from the 22 experts. Firstly, the "Performance Evaluation Index System for Poverty Alleviation through NTFPs" is taken as the decision-making target level of the hierarchical structure model. Then, the first-level evaluation indicators in the evaluation system, including poverty alleviation achievements, sustainability of poverty alleviation and satisfaction with poverty alleviation, are used as the intermediate factor level. Finally, all the secondary evaluation indicators are listed as alternatives. Thus, we obtain a hierarchical model of a performance evaluation index system for poverty alleviation through NTFPs (Figure 3). And we explain the meaning of each indicator element layer (Table 3).

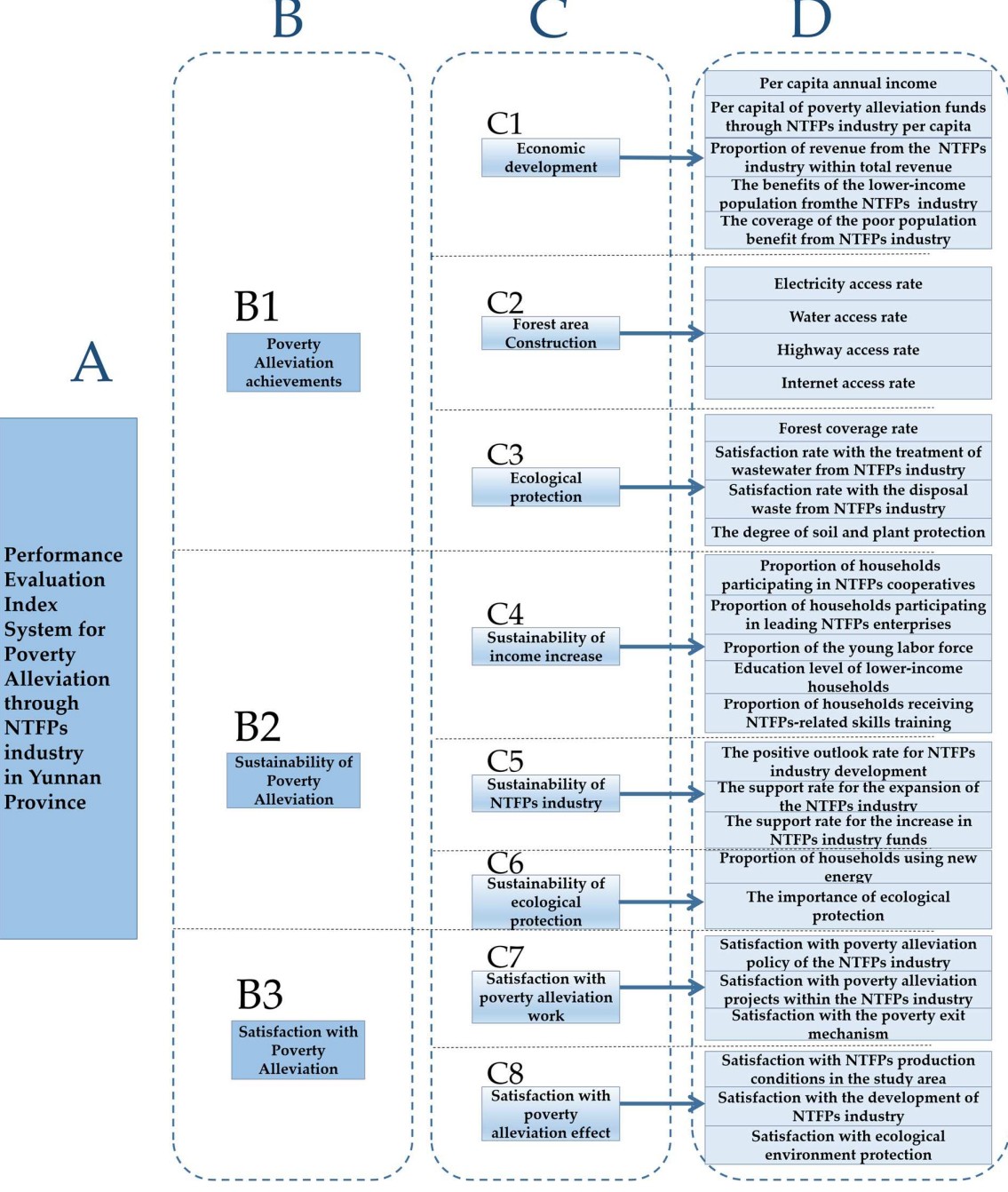

**Figure 3.** The performance evaluation index system for the poverty alleviation performance of the NTFPs industry.

**Table 3.** The interpretation of the indicator element layers.

| Indicator Element Layer | Interpretation |
|---|---|
| Per capita annual income | The average annual income per person, which reflects the economic situation of the household |
| Per capita poverty alleviation funds for the NTFPs industry | Funds provided to lower-income farmers by the government to develop the NTFPs industry |
| Proportion of poverty alleviation through the NTFPs industry | The proportion of lower-income households who achieve poverty alleviation through the NTFPs industry, which can explain the contribution rate of the NTFPs industry to lower-income households |
| Proportion of revenue from the NTFPs industry within total revenue | The proportion of lower-income households who achieve poverty alleviation through the NTFPs industry, which reflects the direct economic benefits of the NTFPs industry for lower-income households |
| The benefits to the lower-income population from the NTFPs industry | The number of lower-income households participating in the NTFPs industry (higher values indicate a stronger driving force of the NTFPs industry on the economy of lower-income households) |
| Electricity access rate<br>Water access rate<br>Highway access rate<br>Internet access rate | The electricity, water, highway and internet access rates in the study area, which indicate improvements in the production conditions of the NTFPs industry |
| Forest coverage rate | Forest coverage rate is an important indicator reflecting the amount of forest resources and forest land occupation in a region, which is an important indicator for the sustainable development of ecological environments |
| Satisfaction rate with the treatment of wastewater from the NTFPs industry<br>Satisfaction rate with the disposal of waste from the NTFPs industry | NTFPs industry development can produce sewage and domestic garbage (the more satisfied lower-income households are with the waste treatment, the lower the impact of the NTFPs industry on the ecological environment) |
| The degree of soil and plant protection | The development of the NTFPs industry affects surrounding soil and plant ecosystems (the degree of soil and plant protection reflects the impact of NTFPs industry development on ecosystems) |
| Proportion of households participating in NTFPs cooperatives<br>Proportion of households participating in leading NTFPs enterprise | Cooperatives and enterprises can integrate resources and provide technology, markets and other services for the development of the NTFPs industry (the higher the proportion of lower-income households participating in cooperatives and enterprises, the stronger their developmental capacity) |
| Proportion of the young labor force | The young labor force is vital for the development of the NTFPs industry (the higher the proportion of the young labor force, the more sustainable the income increase) |
| Education level of lower-income households | The higher the education level of lower-income households, the more positive their thinking, the higher their degree of training and the stronger their developmental ability |
| Proportion of households receiving NTFPs-related skills training | The more that lower-income households receive training related to the NTFPs industry, the more sustainable the poverty alleviation performance of the NFTPs industry |
| The positive outlook rate for NTFPs industry development | Households are the main body of the development of the NTFPs industry (the more optimistic they are about the industry, the more sustainable its development) |
| The support rate for the expansion of the NTFPs industry | Expanding the scale of the NTFPs industry is conducive to the further development of the NTFPs industry and the sustainable poverty alleviation performance of the NTFPs industry |
| The support rate for the increase in NTFPs industry funds | Lower-income households can use their funds for the development of various industries (the higher the proportion of funds used for the development of the NTFPs industry, the more sustainable the NTFPs industry) |
| Proportion of households using new energy | The use of new energy is conducive to ecological environment protection (the proportion of households using new energy reflects the sustainability of ecological protection) |
| The importance of ecological protection | The degree to which lower-income households think ecological protection is important directly reflects the sustainability of ecological protection |

**Table 3.** *Cont.*

| Indicator Element Layer | Interpretation |
|---|---|
| Satisfaction with the poverty alleviation policy of the NTFPs industry | The policy is an important basis and guarantee for carrying out poverty alleviation through the NTFPs industry (the satisfaction of lower-income households with the poverty alleviation policy of the NTFPs industry directly reflects the implementation of the poverty alleviation policy) |
| Satisfaction with poverty alleviation projects within the NTFPs industry | These projects are important for carrying out poverty alleviation through the NTFPs industry (the higher the satisfaction with the projects, the more suitable the projects are for local development and the more able they are to drive lower-income households out of poverty) |
| Satisfaction with the poverty exit mechanism | Whether the exit mechanism is fair and whether lower-income households are satisfied with the exit mechanism are important indicators for measuring satisfaction with poverty alleviation performance |
| Satisfaction with the NTFPs production conditions in the study area | This indicator demonstrates whether the current infrastructure meets the needs of NTFPs industry development in the view of lower-income households (the more satisfied lower-income households are with the production conditions, the more effective the poverty alleviation policy in improving the production conditions) |
| Satisfaction with the development of the NTFPs industry | The development of the NTFPs industry is an important indicator reflecting its poverty alleviation effect (the satisfaction of lower-income households with the development of the NTFPs industry measures their satisfaction with its poverty alleviation effect) |
| Satisfaction with ecological environment protection | The development of the NTFPs industry could not only improve economic situations but could also play a role in protecting the environment (the protection of ecological environments is also a reflection of the poverty alleviation effect) |

### 2.3.2. Constructing the Judgment Matrix

In order to improve the rationality of index weights, Saaty (1980) proposed the consistent matrix method (CMM), which does not compare all factors together but rather compares each factor to each other [38]. Furthermore, a relative scale was used for comparison to minimize the difficulty in comparing factors and improve the accuracy [39]. Then, the experts rated the relative importance (between two indices) of N indicators at the same level, using a 1–9 scale of relative importance (Table 4).

**Table 4.** The standard group values.

| Number | Comparison Values | Meaning |
|---|---|---|
| 1 | 1 | Both are of the same importance |
| 2 | 3 | The former is slightly more important than the latter |
| 3 | 5 | The former is somewhat more important than the latter |
| 4 | 7 | The former is much more important than the latter |
| 5 | 9 | The former is significantly more important than the latter |
| 6 | 2, 4, 6 and 8 | The intermediate values between the above adjacent judgments |

Then, using $a_{ij}$ to express the comparison result of the *i*th factor to the *j*th factor, we obtained judgment matrix A, which was an orthogonal matrix:

$$A = \left(a_{ij}\right)_{n \times n} = \begin{pmatrix} a_{11} & a_{12} & \cdots & a_{1n} \\ a_{21} & a_{22} & \cdots & a_{2n} \\ \vdots & \vdots & \cdots & \vdots \\ a_{n1} & a_{n2} & \cdots & a_{nn} \end{pmatrix} \quad a_{ij} > 0, \ a_{ij} = \frac{1}{a_{ji}}$$

By averaging the scores of the experts, we obtained 12 judgment matrices (Tables A1–A12 in Appendix B). To clarify the results more clearly, the judgment matrix for each indicator element layer in the sustainability of income increase was analyzed as an example (Table 5). In this judgement matrix, $a_{ij}$ was the average score of the 22 experts.

**Table 5.** C4-D: The judgement matrix for the sustainability of income increase.

| | Proportion of Households Participating in NTFPs Cooperatives | Proportion of Households Participating in Leading NTFPs Enterprises | Proportion of the Young Labor Force | Average Education Level of the Young Labor Force | Proportion of Households Receiving Professional Skills Training Related to the NTFPs Industry |
|---|---|---|---|---|---|
| Proportion of households participating in NTFPs cooperatives | 1 | 1.8571 | 1.2857 | 1 | 0.8889 |
| Proportion of households participating in leading NTFPs enterprises | 0.5385 | 1 | 1.3750 | 1.7778 | 1.8000 |
| Proportion of the young labor force | 0.7778 | 0.7273 | 1 | 1.1667 | 1.4000 |
| Average education level of the young labor force | 1 | 0.5625 | 0.8571 | 1 | 1.88889 |
| Proportion of households receiving professional skills training related to the NTFPs industry | 1.1250 | 0.5556 | 0.7143 | 0.5294 | 1 |

### 2.3.3. Calculating the Index Weights

We used the root squaring method to determine the weight of each index. The specific steps were as follows:

(1) Calculate the geometric mean value $W_i^0$ of each line of the judgment matrix using the root squaring method:

$$W_i^0 = \left( \prod_{j=1}^n a_{ij} \right)^{\frac{1}{n}} \qquad i, j = 1, 2, 3, \ldots\ldots n$$

where $a_{ij}$ represents the elements in the $i$th row and the $j$th column of the original judgment matrix, $n$ represents the number of indicators and $W_i^0$ represents the geometric mean of the $i$th row of the original judgment matrix.

(2) Normalize the geometric mean of each line to obtain the respective eigenvectors:

$$W_i = \frac{w_i^0}{\sum_{i=1}^n w_i^0} \qquad i, j = 1, 2, 3, \ldots\ldots n$$

where $W_i$ represents the weight of the $i$th indicator, $n$ represents the number of indicators and $W_i^0$ represents the geometric mean value of the $i$th line of the original judgment matrix.

### 2.3.4. Checking the Consistency of the Judgment Matrix

In order to ensure that the calculated weights were scientific and correct, it was necessary to carry out consistency tests on each judgment matrix. Only weights determined using judgment matrices and consistency tests are considered persuasive and credible [40,41]. The steps to check the consistency of the judgment matrices were as follows:

(1) Calculate the maximum eigenvalue of the judgment matrix:

$$\lambda_{max} = \frac{1}{n} \sum_{i=1}^n \frac{(AW)_i}{W_i}$$

where $n$ is the matrix order and $W_i$ is the weight coefficient value of the desired index.

(2) Calculate the consistency index (*C.I.*):

$$C.I. = \frac{\lambda_{max} - n}{n - 1}$$

(3)    Calculate the consistency ratio (*C.R.*):

$$\frac{C.I.}{R.I.} = C.R.$$

where *R.I.* is the average random consistency index, which is fixed and known. The *R.I.* values are shown in Table 6.

**Table 6.** The average random consistency index (R.I.) values.

| n * | 1 | 2 | 3 | 4 | 5 | 6 | 7 | 8 | 9 | 10 | 11 | 12 | 13 | 14 | 15 |
|-----|---|---|---|---|---|---|---|---|---|----|----|----|----|----|----|
| R.I. | 0 | 0 | 0.52 | 0.89 | 1.12 | 1.26 | 1.36 | 1.41 | 1.46 | 1.49 | 1.52 | 1.54 | 1.56 | 1.58 | 1.59 |

Note: The judgment matrices were constructed using the random method and after more than 500 repeated calculations, the consistency index values were calculated and averaged; * *n* is the dimension of judgement matrix (as shown in Table 3, when there were five indicators in the judgment matrix, i.e., *n* = 5).

When *C.R.* ≤ 0.1, the judgment matrix was considered consistent and the calculated weight could be accepted.

Taking the judgment matrix in Table 3 as an example, we calculated that $\lambda_{max}$ = 5.2276 and *C.I.* = 0.0569. According to Table 4, when *n* = 5, then *R.I.* = 1.12. Then, we could obtain that *C.R.* = 0.0508, which was less than 0.1 and, therefore, this matrix was considered to have satisfactory consistency. This means that pairwise comparisons of 22 experts are consistent [39,42].

In the same way, we checked the judgment matrix between different levels. A–B was the judgment matrix of the target level and first-level evaluation indices, which included the achievement of poverty alleviation, the sustainability of poverty alleviation and satisfaction with poverty alleviation. Bi–C was the judgment matrix of the first-level evaluation indices and the second-level evaluation indices. For example, B1–C was the judgment matrix of economic development, forest area construction and ecological protection. Ci–D was the judgment matrix of the secondary evaluation indices and the specific evaluation indices, meaning that C1–D was the judgment matrix of electricity access rate, water access rate, highway access rate and internet access rate in the study area. As can be seen from Table 7, all judgment matrices passed the consistency tests, which indicated that the calculated weights were scientific and that the results were reliable.

**Table 7.** The consistency test results for the judgment matrices.

| Judgement Matrix | $\lambda_{max}$ | *n* | *C.I.* | *C.R.* |
|------------------|-----------------|-----|--------|--------|
| A–B | 3.0462 | 3 | 0.0231 | 0.0444 |
| B1–C | 3.0308 | 3 | 0.0154 | 0.0296 |
| B2–C | 3.0431 | 3 | 0.0216 | 0.0415 |
| B3–C | 2.0000 | 2 | 0 | 0 |
| C1–D | 5.2985 | 5 | 0.0746 | 0.0666 |
| C2–D | 4.0755 | 4 | 0.0252 | 0.0283 |
| C3–D | 4.0386 | 4 | 0.0129 | 0.0145 |
| C4–D | 5.2276 | 5 | 0.0569 | 0.0508 |
| C5–D | 3.0003 | 3 | 0.0002 | 0.0003 |
| C6–D | 2.0000 | 2 | 0 | 0 |
| C7–D | 3.0047 | 3 | 0.0024 | 0.0045 |
| C8–D | 3.0249 | 3 | 0.0125 | 0.0239 |

*2.4. Comprehensive Evaluation Method*

2.4.1. Determining the Index Reference Values

To evaluate the poverty alleviation performance of the NTFPs industry, it was necessary to determine the reference values of each evaluation index element level. As shown in Table 8, we used four methods to determine the reference values.

**Table 8.** The reference values of each index element level.

| Indicator Element Layer | Unit | Investigation or Calculation Method | Reference Value | Determination Method * |
|---|---|---|---|---|
| Per capita annual income (D11) | CNY | Annual household net income/Permanent household population | CNY 2952 | A |
| Per capita poverty alleviation funds for the NTFPs industry (D12) | CNY | Poverty alleviation funds for the NTFPs industry/Total lower-income population | CNY 1193.52 | A |
| Proportion of poverty alleviation through the NTFPs industry (D13) | % | Population lifted out of poverty through the NTFPs industry/Total population lifted out of poverty × 100% | 26.5% | A |
| Proportion of revenue from the NTFPs industry within total revenue (D14) | % | NTFPs industry revenue/Total revenue × 100% | 3.5% | C |
| The benefits to the lower-income population from the NTFPs industry (D15) | % | Population lifted out of poverty through the NTFPs industry/Total lower-income population × 100% | 15% | A |
| Electricity access rate (D21) | % | Number of households with access to electricity/Total number of lower-income households × 100% | 100% | B |
| Water access rate (D22) | % | Number of households with access to water/Total number of lower-income households × 100% | 100% | B |
| Highway access rate (D23) | % | Number of households with access to highways/Total number of lower-income households × 100% | 100% | B |
| Internet access rate (D24) | % | Number of households with internet access/Total number of lower-income households × 100% | 100% | B |
| Forest coverage rate (D31) | % | Area of forest/Total land area × 100% | 78.98% | C |
| Satisfaction rate with the treatment of wastewater from the NTFPs industry (D32) | % | Number of lower-income households satisfied with the treatment of wastewater from the NTFPs industry/Total number of lower-income households × 100% | 100% | B |
| Satisfaction rate with the disposal of waste from the NTFPs industry (D33) | % | Number of lower-income households satisfied with the disposal of waste from the NTFPs industry/Total number of lower-income households × 100% | 100% | B |
| The degree of soil and plant protection (D34) | % | Number of lower-income households satisfied with soil and plant protection/Total number of lower-income households × 100% | 100% | B |
| The proportion of households participating in NTFPs cooperatives (D41) | % | Number of households participating in NTFPs cooperatives/Total number of lower-income households × 100% | 100% | B |
| The proportion of households participating in leading NTFPs enterprises (D42) | % | Number of households participating in leading NTFPs enterprises/Total number of lower-income households × 100% | 100% | B |
| Proportion of the young labor force (D43) | % | Labor force aged 16–44/Total labor force × 100% | 80% | D |
| Education level of lower-income households (D44) | Years | Total number of years in education of lower-income households/Total number of lower-income households | 7.06 | C |
| Proportion of households receiving NTFPs-related skills training (D45) | % | Number of households receiving NTFPs-related skills training/Total number of lower-income households × 100% | 100% | B |
| The positive outlook rate for NTFPs industry development (D51) | % | Number of households with a positive outlook on the NTFPs industry/Total number of lower-income households × 100% | 100% | B |
| The support rate for the expansion of the NTFPs industry (D52) | % | Number of households supporting the expansion of the NTFPs industry/Total number of lower-income households × 100% | 100% | B |
| The support rate for the increase in NTFPs industry funds (D53) | % | Funds for developing the NTFPs industry/Total funds for developing all industries × 100% | 100% | B |
| Proportion of households using new energy (D61) | % | Number of households using new energy/Total number of lower-income households × 100% | 100% | B |
| The importance of ecological protection (D62) | % | Number of lower-income households willing to invest time and money in ecological protection/Total number of lower-income households × 100% | 100% | B |
| Satisfaction with the poverty alleviation policy of the NTFPs industry (D71) | % | Number of households satisfied with the poverty alleviation policy of the NTFPs industry/Total number of lower-income households × 100% | 100% | B |

**Table 8.** *Cont.*

| Indicator Element Layer | Unit | Investigation or Calculation Method | Reference Value | Determination Method * |
|---|---|---|---|---|
| Satisfaction with poverty alleviation projects within the NTFPs industry (D72) | % | Number of households satisfied with poverty alleviation projects within the NTFPs industry/Total number of lower-income households × 100% | 100% | B |
| Satisfaction with the poverty exit mechanism (D73) | % | Number of households satisfied with the poverty exit mechanism/Total number of lower-income households × 100% | 100% | B |
| Satisfaction with the NTFPs production conditions in the study area (D81) | % | Number of households satisfied with the NTFPs production conditions in the study area/Total number of lower-income households × 100% | 100% | B |
| Satisfaction with the development of the NTFPs industry (D82) | % | Number of households satisfied with the development of the NTFPs industry/Total number of lower-income households × 100% | 100% | B |
| Satisfaction with ecological environment protection (D83) | % | Number of households satisfied with their ecological environment/Total number of lower-income households × 100% | 100% | B |

Note: * Method A was based on planning objectives that were clearly formulated by poverty alleviation or forestry authorities; Method B was used for theoretically optimal values; Method C used local averages for economic, social and ecological development indicators; Method D was proposed in the National Agricultural Modernization Development Level Report (2016).

Method A was based on the planning objectives that have been clearly formulated by poverty alleviation or forestry authorities, such as per capita annual income. Based on the 2016 national poverty exit mechanism for determining whether people have been alleviated from poverty, the poverty line was CNY 2952 per capita; therefore, CNY 2952 was taken as the reference index value.

Method B was used for theoretically optimal values, such as the electricity access rate in the study area. In order to develop the NTFPs industry, the electricity access rate in the area should be 100%.

Method C was used for local or national average values, such as the forest coverage rate. The local forest coverage rate in Lushui City is 78.98%; therefore, 78.98% was taken as the reference index value.

Method D was proposed in the National Agricultural Modernization Development Level Report (2016), which pointed out that considering the nine-year compulsory education period and the current status of the young workforce in rural areas, the target value for the young labor force in rural areas is 80%.

2.4.2. Calculating the Index Scores

Firstly, we needed to calculate the specific scores for each indicator. By comparing the actual scores and the weights of each indicator, we could obtain the specific scores. When the actual score was more than the weight, the indicator was valid and vice versa. The specific calculation method for the actual index scores was as follows:

$$S_i = \frac{A}{R} \times W \times 100\%$$

where $A$ is the actual score from the questionnaire, $R$ is the reference score from Table 6 and $W$ is the weight of the index.

Then, we could obtain the total score for the poverty alleviation performance of the NTFPs industry by summing the scores of all indicators. Based on previous studies, we used a five-level classification method to evaluate the scores [43,44]. When the final actual evaluation score fell in the interval of [0, 60), it demonstrated poor poverty alleviation performance; scores in the interval of [60, 70) demonstrated acceptable poverty alleviation performance; scores in the interval of [70, 80) demonstrated good poverty alleviation performance; scores in the interval of [80, 90) demonstrated very good alleviation performance; scores in the interval of [90, 100] demonstrated excellent poverty alleviation performance.

## 3. Results

### 3.1. Basic Information of the Households

In the survey, a total of 150 questionnaires were collected. After eliminating those missing values for the relevant data, 139 valid samples were obtained. Among the valid questionnaires, the proportions of male and female respondents were 53.2% and 46.8%, respectively. The age of respondents was concentrated between 18 and 65 years old, accounting for 57.6%. In terms of ethnicity, more than half of the respondents were Lisu. Additionally, the education levels of the surveyed farmers were generally low. As shown in Table 9, farmers with a middle school education accounted for 59% of respondents and only 10.8% of farmers had a high school-level education. In addition, 97.8% of the surveyed farmers had medical insurance and 89.2% had endowment insurance.

**Table 9.** The basic information of the participating households.

| Characteristic | Category | Percentage |
|---|---|---|
| Gender | Male | 53.2% |
| | Female | 46.8% |
| Age * | 18–65 years old | 57.6% |
| | 66–79 years old | 23.8% |
| | Over 80 years old | 18.6% |
| Ethnicity | Han | 39.6% |
| | Lisu | 58.3% |
| | Other | 2.1% |
| Education level | Did not attend school | 4.6% |
| | Primary school (Grades 1–6) | 25.6% |
| | Middle school (Grades 7–9) | 59.0% |
| | High school (Grades 10–) | 10.8% |
| Social security ** | Medical insurance | 97.8% |
| | Endowment insurance | 89.2% |

Note: * In 2022, the United Nations World Health Organization made a new age division, 0–17 years old for minors, 18–65 years old for youth, 66–79 years old for middle age, 80–99 years old for the elderly, over 100 years old for longevity. ** Medical insurance is a medical mutual aid system through government organization, guidance and support, with voluntary participation from farmers. Endowment insurance is a kind of social insurance system. According to certain laws and regulations, the system is established by the government to guarantee the basic livelihood of elderly people who have lost their ability to work. These are two different types of insurance that guarantee the livelihood of farmers, and farmers can choose both of them.

### 3.2. The Performance Evaluation Index System

By establishing a judgment matrix for each level, calculating the weights of each index and testing the consistency of the judgment matrices, we developed the performance evaluation index system for determining the poverty alleviation performance of the NTFPs industry in Yunnan Province (Table 10).

**Table 10.** The performance evaluation index system for the poverty alleviation performance of the NTFPs industry in Yunnan Province.

| Criterion Layer | Weight | Indicator Layer | Weight | Indicator Element Layer | Weight |
|---|---|---|---|---|---|
| The achievement of poverty alleviation (B1) | 0.5128 | Economic development (C1) | 0.2479 | Per capita annual income (D11) | 0.0925 |
| | | | | Per capita poverty alleviation funds for the NTFPs industry (D12) | 0.0423 |
| | | | | Proportion of poverty alleviation through the NTFPs industry (D13) | 0.0587 |
| | | | | Proportion of revenue from the NTFPs industry within total revenue (D14) | 0.0289 |
| | | | | The benefits to the lower-income population from the NTFPs industry (D15) | 0.0255 |
| | | Forest area construction (C2) | 0.1209 | Electricity access rate (D21) | 0.0452 |
| | | | | Water access rate (D22) | 0.0348 |
| | | | | Highway access rate (D23) | 0.0277 |
| | | | | Internet access rate (D24) | 0.0132 |
| | | Ecological protection (C3) | 0.1440 | Forest coverage rate (D31) | 0.0537 |
| | | | | Satisfaction rate with the treatment of wastewater from the NTFPs industry (D32) | 0.0398 |
| | | | | Satisfaction rate with the disposal of waste from the NTFPs industry (D33) | 0.0207 |
| | | | | The degree of soil and plant protection (D34) | 0.0298 |

**Table 10.** *Cont.*

| Criterion Layer | Weight | Indicator Layer | Weight | Indicator Element Layer | Weight |
|---|---|---|---|---|---|
| The sustainability of poverty alleviation (B2) | 0.3631 | Sustainability of income increase (C4) | 0.2140 | Proportion of households participating in NTFPs cooperatives (D41) | 0.0507 |
| | | | | Proportion of households participating in leading NTFPs enterprises (D42) | 0.0503 |
| | | | | Proportion of the young labor force (D43) | 0.0399 |
| | | | | Education level of lower-income households (D44) | 0.0411 |
| | | | | Proportion of households receiving NTFPs-related skills training (D45) | 0.0320 |
| | | Sustainability of the NTFPs industry (C5) | 0.0718 | The positive outlook rate for NTFPs industry development (D51) | 0.0277 |
| | | | | The support rate for the expansion of the NTFPs industry (D52) | 0.0218 |
| | | | | The support rate for the increase in NTFPs industry funds (D53) | 0.0223 |
| | | Sustainability of ecological protection (C6) | 0.0773 | Proportion of households using new energy (D61) | 0.0505 |
| | | | | The importance of ecological protection (D62) | 0.0268 |
| Satisfaction with poverty alleviation (B3) | 0.1241 | Satisfaction with poverty alleviation work (C7) | 0.0564 | Satisfaction with the poverty alleviation policy of the NTFPs industry (D71) | 0.0239 |
| | | | | Satisfaction with poverty alleviation projects within the NTFPs industry (D72) | 0.0219 |
| | | | | Satisfaction with the poverty exit mechanism (D73) | 0.0106 |
| | | Satisfaction with poverty alleviation effect (C8) | 0.0677 | Satisfaction with the NTFPs production conditions in the study area (D81) | 0.0312 |
| | | | | Satisfaction with the development of the NTFPs industry (D82) | 0.0265 |
| | | | | Satisfaction with ecological environment protection (D83) | 0.0100 |

### 3.3. Poverty Alleviation Performance of the NTFPs Industry

According to the questionnaires and interviews, we evaluated the poverty alleviation performance of the NTFPs industry in Sanhe Village using a comprehensive evaluation method. In order to express the evaluation results using a 100-point system, the weights were replaced with a 100-point system. As shown in Table 11, the final poverty alleviation performance score was 79.33, which indicated that the poverty alleviation performance of the NTFPs industry in Sanhe Village was good.

**Table 11.** The performance evaluation index system for the poverty alleviation performance of the NTFPs industry.

| Criterion Layer | Indicator Layer | Indicator Element Layer | Reference Value | Actual Value | Weight (%) | Actual Score |
|---|---|---|---|---|---|---|
| The achievement of poverty alleviation (B1) | Economic development (C1) | Per capita annual income (D11) | 2952 | 2958.62 | 9.25 | 9.27 |
| | | Per capita poverty alleviation funds for the NTFPs industry (D12) | 1193.52 | 862.57 | 4.23 | 3.06 |
| | | Proportion of poverty alleviation through the NTFPs industry (D13) | 26.50% | 34.28% | 5.87 | 7.59 |
| | | Proportion of revenue from the NTFPs industry within total revenue (D14) | 3.50% | 4.60% | 2.89 | 3.80 |
| | | The benefits to the lower-income population from the NTFPs industry (D15) | 15% | 17.50% | 2.55 | 2.98 |
| | | | | | 24.79 | 26.7 |
| | Forest area construction (C2) | Electricity access rate (D21) | 100% | 95% | 4.52 | 4.29 |
| | | Water access rate (D22) | 100% | 80% | 3.48 | 2.78 |
| | | Highway access rate (D23) | 100% | 85% | 2.77 | 2.35 |
| | | Internet access rate (D24) | 100% | 65% | 1.32 | 0.86 |
| | | | | | 12.09 | 10.28 |
| | Ecological protection (C3) | Forest coverage rate (D31) | 78.98% | 92.00% | 5.37 | 6.26 |
| | | Satisfaction rate with the treatment of wastewater from the NTFPs industry (D32) | 100% | 85.50% | 3.98 | 3.40 |
| | | Satisfaction rate with the disposal of waste from the NTFPs industry (D33) | 100% | 70.60% | 2.07 | 1.46 |
| | | The degree of soil and plant protection (D34) | 100% | 82.70% | 2.98 | 2.46 |
| | | | | | 14.4 | 13.58 |

**Table 11.** *Cont.*

| Criterion Layer | Indicator Layer | Indicator Element Layer | Reference Value | Actual Value | Weight (%) | Actual Score |
|---|---|---|---|---|---|---|
| The sustainability of poverty alleviation (B2) | Sustainability of income increase (C4) | Proportion of households participating in NTFPs cooperatives (D41) | 100% | 70.80% | 5.07 | 3.59 |
| | | Proportion of households participating in leading NTFPs enterprises (D42) | 100% | 12.50% | 5.03 | 0.63 |
| | | Proportion of the young labor force (D43) | 80% | 46.80% | 3.99 | 2.33 |
| | | Education level of lower-income households (D44) | 7.06 | 5.90 | 4.11 | 3.43 |
| | | Proportion of households receiving NTFPs-related skills training (D45) | 100% | 68.50% | 3.20 | 2.19 |
| | | | | | 21.4 | 12.17 |
| | Sustainability of the NTFPs industry (C5) | The positive outlook rate for NTFPs industry development (D51) | 100% | 45.40% | 2.77 | 1.26 |
| | | The support rate for the expansion of the NTFPs industry (D52) | 100% | 61.90% | 2.18 | 1.35 |
| | | The support rate for the increase in NTFPs industry funds (D53) | 100% | 60.80% | 2.23 | 1.36 |
| | | | | | 7.18 | 3.97 |
| | Sustainability of ecological construction (C6) | Proportion of households using new energy (D61) | 100% | 9.80% | 5.05 | 0.49 |
| | | The importance of ecological protection (D62) | 100% | 72.50% | 2.68 | 1.94 |
| | | | | | 7.73 | 2.43 |
| Satisfaction with poverty alleviation (B3) | Satisfaction with poverty alleviation work (C7) | Satisfaction with the poverty alleviation policy of the NTFPs industry (D71) | 100% | 80.50% | 2.39 | 1.92 |
| | | Satisfaction with poverty alleviation projects within the NTFPs industry (D72) | 100% | 75.50% | 2.19 | 1.65 |
| | | Satisfaction with the poverty exit mechanism (D73) | 100% | 85.50% | 1.06 | 0.91 |
| | | | | | 5.64 | 4.48 |
| | Satisfaction with poverty alleviation effect (C8) | Satisfaction with the NTFPs production conditions in the study area (D81) | 100% | 78.50% | 3.12 | 2.45 |
| | | Satisfaction with the development of the NTFPs industry (D82) | 100% | 90.50% | 2.65 | 2.40 |
| | | Satisfaction with ecological environment protection (D83) | 100% | 86.50% | 1.00 | 0.87 |
| | | | | | 6.77 | 5.72 |
| Final score | | | | | | 79.33 |

## 4. Discussion

### 4.1. Analysis of the Performance Evaluation Index System

The AHP is often used to determine the weight of index systems, and has been widely used in medicine, management, decision making and other fields [45–48]. At the same time, it is also used in sustainable forest management [49]. Therefore, we use the AHP to determine the weights of the performance evaluation index system for determining the level of poverty alleviation of NTFPs.

The indicator weights could theoretically explain the importance of the indicators in the target layer. In order to clarify the role of each evaluation indicator in the poverty alleviation performance of the NTFPs industry, the indicators were sorted according to their weights (Table 12). Some of the economic indicators, such as income, could most directly reflect the livelihoods of farmers. At the same time, these indicators played significant roles in the poverty alleviation performance of the NTFPs industry. Therefore, we concluded that it would be essential to enhance the income of farmers through the development of the NTFPs industry. The NTFPs industry follows an industrial development model that combines economic and ecological benefits. To evaluate the poverty alleviation effect of the NTFPs industry, it was not only necessary to evaluate the economic benefits but also the effectiveness of the protection of forest resources. We used some indicators, such as forest coverage rate and the proportion of households using new energy, to effectively reflect the effectiveness of the protection of forest resources. Additionally, the sustainability of farmers' incomes and industrial development were also important indicators for evaluating poverty alleviation performance. Professional organizations, such as enterprises and cooperatives, could improve the development capacity of farmers. Farmers could also strengthen their sustainable development capacity by joining these organizations. Therefore, the proportion of households participating in NTFPs enterprises or cooperatives reflected the sustainable

developmental ability of the farmers. It is known that the construction of infrastructures in forest areas is closely related to the sustainable development of the NTFPs industry. Some indicators, such as electricity, water and highway access rates, were used to reflect the degree of infrastructure construction in the study area, which helped us to evaluate the effectiveness of the sustainable development of the NTFPs industry. To sum up, the indicator system we established was consistent with the poverty alleviation goals of the NTFPs industry. Hence, the performance evaluation index system for determining the poverty alleviation performance of the NTFPs industry in Yunnan Province that we constructed using the AHP was in line with the actual situation.

**Table 12.** The ranking of each indicator weight at the indicator element level.

| Indicator Element Layer | Weight |
|---|---|
| Per capita annual income (D11) | 0.0925 |
| The proportion of poverty alleviation through the NTFPs industry (D13) | 0.0587 |
| Forest coverage rate (D31) | 0.0537 |
| Proportion of households participating in NTFPs cooperatives (D41) | 0.0507 |
| Proportion of households using new energy (D61) | 0.0505 |
| Proportion of households participating in leading NTFPs enterprises (D42) | 0.0503 |
| Electricity access rate (D21) | 0.0452 |
| Per capita poverty alleviation funds for the NTFPs industry (D12) | 0.0423 |
| Education level of lower-income households (D44) | 0.0411 |
| Proportion of the young labor force (D43) | 0.0399 |
| Satisfaction rate with the treatment of wastewater from the NTFPs industry (D32) | 0.0398 |
| Water access rate (D22) | 0.0348 |
| Proportion of households receiving NTFPs-related skills training (D45) | 0.0320 |
| Satisfaction with the NTFPs production conditions in the study area (D81) | 0.0312 |
| The degree of soil and plant protection (D34) | 0.0298 |
| Proportion of revenue from the NTFPs industry within total revenue (D14) | 0.0289 |
| Highway access rate (D23) | 0.0277 |
| The positive outlook rate for NTFPs industry development (D51) | 0.0277 |
| The importance of ecological protection (D62) | 0.0268 |
| Satisfaction with the development of the NTFPs industry (D82) | 0.0265 |
| The benefits to the lower-income population from the NTFPs industry (D15) | 0.0255 |
| Satisfaction with the poverty alleviation policy of the NTFPs industry (D71) | 0.0239 |
| The support rate for the increase in NTFPs industry funds (D53) | 0.0223 |
| Satisfaction with poverty alleviation projects within the NTFPs industry (D72) | 0.0219 |
| The support rate for the expansion of the NTFPs industry (D52) | 0.0218 |
| Satisfaction rate with the disposal of waste from the NTFPs industry (D33) | 0.0207 |
| Internet access rate (D24) | 0.0132 |
| Satisfaction with the poverty exit mechanism (D73) | 0.0106 |
| Satisfaction with ecological environment protection (D83) | 0.0100 |

### 4.2. Analysis of Poverty Alleviation Performance

The development of the NTFPs industry in other regions has effectively sustained the livelihoods of local farmers and the quality of the surrounding ecological environments [19,23,50–52]. For a great number of rural (and also urban) inhabitants, particularly the poorest sectors, their use represents an important source of subsistence and income generation [53,54]. Silva et al. (2020) found that NTFPs are a consolidated source of income and acquisition of inputs from forest environments in Brazil [55]. Shackleton et al. (2004) found that despite the small cash incomes from trade, NTFPs provide an important contribution that complements the diverse livelihood strategies within a household, especially for the poorer sectors of rural society in South Africa [56]. Han Feng (2015) found that the development of the NTFPs industry had a significant impact on the economic income of forest households, according to survey results from 368 households in Jiangxi, Fujian, Hunan, Jilin, Chongqing and Shaanxi [57]. Through questionnaires and interviews with farmers in southern Shaanxi, Bai Hui (2021) found that the development of the NTFPs industry had the highest contribution to the economic income of farmers compared to other poverty alleviation methods [58]. Shen Yingying (2021) found that the NTFPs industry could effectively improve the economic income of forest households, according to in-depth research and analysis in Guizhou Province [59]. In general, the NTFPs industry can effectively improve farmers' livelihoods through efficient production, large-scale operations, continuous improvements in product value and the pro-

motion of industrial integration. Similarly, the poverty alleviation performance of the NTFPs industry in Sanhe Village was good, as shown in Table 11. Further, we evaluated the poverty alleviation performance of the NTFPs industry in Sanhe Village from three perspectives: the achievement of poverty alleviation, the sustainability of poverty alleviation and satisfaction with poverty alleviation.

### 4.2.1. The Achievement of Poverty Alleviation

It was obvious that the development of the NTFPs industry in Sanhe Village brought considerable economic income to the local lower-income households. Furthermore, it also played a positive role in forest area construction and ecological environment protection. For example, the lower-income households that were interviewed had an average annual income of CNY 2958.62 from the NTFPs industry, which was above the poverty line set by the state at that time. At the same time, the benefits to the lower-income population from the NTFPs industry also exceeded the expected planning targets. In terms of forest area construction, the NTFPs industry's development also helped to increase electricity, water and highway access rates. However, the internet access rate, which is very important for increasing knowledge, still needs to be improved. In addition, forest resources were effectively protected and reasonably utilized by the NTFPs industry. On the one hand, the forest coverage rate of Sanhe Village was significantly higher than that of Lushui City; on the other hand, farmers used some measures for the treatment of wastewater and the disposal of waste from the NTFPs industry so as to minimize its impact on the ecological environment. In summary, the NTFPs industry belongs to the ecological industry, which has a small impact on ecological environments. In poverty-stricken areas, especially poverty-stricken forest areas, vigorously developing the NTFPs industry could not only help to improve the local economy but also protect the local ecological environment.

### 4.2.2. The Sustainability of Poverty Alleviation

Compared to the ideal scores for the indicators at all levels, we found that there were still many shortcomings in the sustainability of the poverty alleviation performance of the NTFPs industry. Although it would be unrealistic for all farmers to participate in enterprises, the low participation rate seriously affected the increase in income for lower-income households. The proportion of lower-income households participating in NTFPs cooperatives in Sanhe Village was limited and the number of lower-income households participating in NTFPs enterprises was even less. Additionally, it could be seen that the lower-income households were seriously limited in terms of their young labor force, which was also the main factor hindering the sustainable economic growth of the lower-income households. According to a report by the Lushui City government, the average number of years in education per capita in the city is 7.06 years, while in the lower-income households that we interviewed, the average was only 5.9 years, showing the gap between the average education levels in the city and those in forest areas. At the same time, because of some concerns about the products, including unstable market prices and difficult sales situations, some households were not optimistic about the development of the NTFPs industry, which affected its sustainable development. In terms of the sustainability of ecological construction, the proportion of lower-income households using new energy was only 9.8%, which was far from the target value. Therefore, continuing to promote the use of new energy could be an important approach to strengthen the sustainability of ecological construction.

### 4.2.3. Satisfaction with Poverty Alleviation

In general, the lower-income households were satisfied with the work and effect of the poverty alleviation performance of the NTFPs industry. It could be seen that the NTFPs industry played an important role in promoting local poverty alleviation, which also reflected the great contributions made by local governments. It is worth mentioning that 80.5% of the lower-income households were satisfied with the poverty alleviation policy, while 90.5% were satisfied with the development of the NTFPs industry. This indicated

that farmers benefited from the poverty alleviation performance of the NTFPs industry, which in turn encouraged the farmers to actively develop the NTFPs industry. At the same time, 86.5% of the farmers were relatively satisfied with current ecological environment protection. This indicated that the development of the NTFPs industry also played an important role in ecological protection. However, fewer farmers were satisfied with the poverty alleviation projects within the NTFPs industry and the production conditions in the study area. This showed that the poverty alleviation projects within the NTFPs industry and the production conditions in the study area need to be improved. The reason for this is that the poverty alleviation projects in different places are similar and lack overall planning and specific industrial layouts, which can lead to the failure of the projects. Additionally, it is difficult to predict the market situation of the NTFPs industry. For example, prices can fluctuate greatly and specific information can be difficult to grasp.

In this study, we constructed a performance evaluation index system for determining the poverty alleviation performance of the non-timber forest products industry and conducted an empirical analysis of the role of the NTFPs industry in sustaining farmers' livelihoods and protecting ecological environments. This paper could provide a reference for other regions to evaluate the development of the NTFPs industry. However, there were some shortcomings in this study, such as a lack of an in-depth data analysis to explore the internal mechanisms of the poverty alleviation performance of the NTFPs industry. Additionally, due to limited time, the empirical analysis only used Sanhe Village as an example, resulting in a lack of a comparative analysis. At the same time, most of the surveyed households lived in remote areas, with inconvenient transportation and relatively scattered residences, resulting in the insufficient sample size of this study. Therefore, further research should be conducted on the following aspects: the scope of the survey should be expanded and the accuracy of the survey data should be enhanced to provide a better evaluation of the poverty alleviation performance of the NTFPs industry; an in-depth scientific analysis of the data should be conducted to explore the underlying impact mechanisms; finally, in order to enhance the rationality of the performance evaluation index system, a more objective method should be used to determine the indices and their weights.

## 5. Conclusions and Implications

In this paper, we evaluated the effects of the NTFPs industry on improvements in farmers' livelihoods and ecological protection in Sanhe Village using a questionnaire, the analytic hierarchy process and a comprehensive evaluation method. We found that the development of the NTFPs industry in Sanhe Village played important roles in sustaining local farmers' livelihoods and protecting the local ecological environment. In particular, it achieved remarkable results in increasing farmers' incomes, improving infrastructure construction and protecting forest resources. It was obvious that the respondents were satisfied with the poverty alleviation work of the NTFPs industry and its effect. However, due to the lack of business entities, such as enterprises and cooperatives, and the lack of willingness to expand the scale of the NTFPs industry, the sustainability of the farmers' incomes and NTFPs industry development was insufficient. Finally, based on our analysis and evaluation results in combination with data from the forestry resource endowment and development foundation, we devised suggestions that could improve the poverty alleviation performance of the NTFPs industry. Firstly, the government should increase investment and attract social capital to support the development of the NTFPs industry, especially in lower-income forest areas. At the same time, relevant departments should deepen the reform of forest property mortgage loans. Secondly, the relevant departments should strengthen the publicity of NTFPs cooperatives and encourage more farmers to voluntarily participate in those cooperatives. Thirdly, the government should introduce high-quality enterprises to maximize the development capacity of lower-income households. Fourthly, targeted training activities should be carried out to improve the knowledge of lower-income households, thereby improving their development capabilities. Finally, the government should plan and cultivate different development models

based on local resources, the willingness of local households to participate and the local need for ecological protection.

**Author Contributions:** Conceptualization, Y.D. and J.W.; methodology, Y.L. and X.C.; validation, Y.D. and X.Z.; formal analysis, Y.D. and J.W.; investigation, Y.D. and Y.L.; writing—original draft preparation, Y.D. and J.W.; writing—review and editing, X.C. and X.Z.; supervision, X.C. All authors have read and agreed to the published version of the manuscript.

**Funding:** This research was funded by the Land greening and ecological restoration management projects of the National Forestry and Grassland Administration "Vegetation suitability evaluation in the Yellow River basin", funding number is 2130205.

**Informed Consent Statement:** Informed consent was obtained from all the individual participants included in the study.

**Data Availability Statement:** The data presented in this study are available on request from the corresponding author.

**Acknowledgments:** All authors gratefully acknowledge the support of the People's Government of Sanhe Village that participated in the investigation, especially the forest households who participated in the questionnaire survey. Moreover, we also gratefully acknowledge 30 relevant personnel for their suggestions on the determination of the indicator system.

**Conflicts of Interest:** The authors declare no conflict of interest.

## Appendix A

*Appendix A.1. Outline of Interviews*

Appendix A.1.1. Outline of Interviews with Forestry Departments at All Levels

(1) Basic information: forestry resources (forest coverage, forest area, etc.); types, distribution and scale of NTFPs industry;
(2) Development status: yield of products NTFPs; management benefits (economic benefits, ecological benefits and social benefits);
(3) Poverty alleviation: the feasibility and importance of poverty alleviation through NTFPs industry; the advantages, disadvantages, opportunities and challenges of poverty alleviation through NTFPs industry; the mode, effect and problems of poverty alleviation through NTFPs industry.

Appendix A.1.2. Outline of Interviews with Poverty Alleviation Offices at All Levels

(1) Poverty profile: distribution and basic situation of poor counties (villages); poor households;
(2) Poverty alleviation projects: the types, proportion and benefits of poverty alleviation through NTFPs industry;
(3) Poverty alleviation: poverty alleviation policies and implementation; farmers' participation; enterprise poverty alleviation efforts;
(4) Poverty alleviation evaluation: achievements, problems, difficulties and improvement measures.

Appendix A.1.3. Outline of Interviews with Village Cadres in Poverty-Stricken Counties

(1) Basic information: number of households (total number of households, number of poor households); population (total population, number of poor people); area (total area, forest land area, cultivated land area, etc.); economic income; income composition; industrial structure;
(2) Forest resources (main tree species types of economic forests, area, proportion and subsidies of public welfare forests); the development status of NTFPs industry; the development of cooperatives and enterprises about NTFPs industry.

## Appendix B.

*Judgment Matrix*

**Table A1.** A–B: Judgement matrix of the poverty alleviation performance through NTFPs.

|  | Poverty Alleviation Achievements | Sustainability of Poverty Alleviation | Satisfaction with Poverty Alleviation |
|---|---|---|---|
| Poverty Alleviation achievements | 1.0000 | 1.7500 | 3.3333 |
| Sustainability of Poverty Alleviation | 0.5714 | 1.0000 | 3.6250 |
| Satisfaction with Poverty Alleviation | 0.3000 | 0.2759 | 1.0000 |

**Table A2.** B1–C: Judgement matrix of the poverty alleviation achievements.

|  | Economic Development | Forest Area Construction | Ecological Protection |
|---|---|---|---|
| Economic development | 1.0000 | 2.4444 | 1.4444 |
| Forest area Construction | 0.4091 | 1.0000 | 1.0000 |
| Ecological protection | 0.6923 | 1.0000 | 1.0000 |

**Table A3.** B2–C: Judgement matrix of the sustainability of poverty alleviation.

|  | Sustainability of Income Increase | Sustainability of NTFPs Industry | Sustainability of Ecological Protection |
|---|---|---|---|
| Sustainability of income increase | 1.0000 | 3.6667 | 2.2500 |
| Sustainability of NTFPs industry | 0.2727 | 1.0000 | 1.1429 |
| Sustainability of ecological protection | 0.4444 | 0.8750 | 1.0000 |

**Table A4.** B3–C: Judgement matrix of the satisfaction with poverty alleviation.

|  | Satisfaction with Poverty Alleviation Work | Satisfaction with Poverty Alleviation Effect |
|---|---|---|
| Satisfaction with poverty alleviation work | 1.0000 | 0.8333 |
| Satisfaction with poverty alleviation effect | 1.2000 | 1.0000 |

**Table A5.** C1–D: Judgement matrix of the economic development.

|  | Per Capita Annual Income | Per Capita Poverty Alleviation Funds for the NTFPs Industry | Proportion of Poverty Alleviation through the NTFPs Industry | Proportion of Revenue from the NTFPs Industry within Total Revenue | The Benefits to the Lower-Income Population from the NTFPs Industry |
|---|---|---|---|---|---|
| Per capita annual income | 1.0000 | 2.4000 | 2.4444 | 3.0000 | 2.6250 |
| Per capita poverty alleviation funds for the NTFPs industry | 0.4167 | 1.0000 | 2.6000 | 2.3333 | 1.4286 |
| Proportion of poverty alleviation through the NTFPs industry | 0.4091 | 0.3846 | 1.0000 | 2.1111 | 2.3750 |
| Proportion of revenue from the NTFPs industry within total revenue | 0.3333 | 0.4286 | 0.4737 | 1.0000 | 1.88889 |
| The benefits to the lower-income population from the NTFPs industry | 0.3810 | 0.7000 | 0.4211 | 0.5294 | 1.0000 |

**Table A6.** C2–D: Judgement matrix of the forest area construction.

| | Electricity Access Rate | Water Access Rate | Highway Access Rate | Internet Access Rate |
|---|---|---|---|---|
| Electricity access rate | 1.0000 | 1.625 | 1.7143 | 2.6667 |
| Water access rate | 0.6154 | 1.0000 | 1.6667 | 2.5000 |
| Highway access rate | 0.5833 | 0.6000 | 1.0000 | 2.8889 |
| Internet access rate | 0.3750 | 0.4000 | 0.3462 | 1.0000 |

**Table A7.** C3–D: Judgement matrix of the ecological protection.

| | Forest Coverage Rate | Satisfaction Rate with the Treatment of Wastewater from the NTFPs Industry | Satisfaction Rate with the Disposal of Waste from the NTFPs Industry | The Degree of Soil and Plant Protection |
|---|---|---|---|---|
| Forest coverage rate | 1.0000 | 1.7500 | 2.1111 | 1.6667 |
| Satisfaction rate with the treatment of wastewater from the NTFPs industry | 0.5714 | 1.0000 | 2.2857 | 1.4444 |
| Satisfaction rate with the disposal of waste from the NTFPs industry | 0.4737 | 0.4375 | 1.0000 | 0.6667 |
| The degree of soil and plant protection | 0.6000 | 0.6923 | 1.5000 | 1.0000 |

**Table A8.** C4–D: Judgement matrix of the sustainability of income increase.

| | Proportion of Households Participating in NTFPs Cooperatives | Proportion of Households Participating in Leading NTFPs Enterprises | Proportion of the Young Labor Force | Education level of Lower-Income Households | Proportion of Households Receiving NTFPs-Related Skills Training |
|---|---|---|---|---|---|
| Proportion of households participating in NTFPs cooperatives | 1 | 1.8571 | 1.2857 | 1 | 0.8889 |
| Proportion of households participating in leading NTFPs enterprises | 0.5385 | 1 | 1.3750 | 1.7778 | 1.8000 |
| Proportion of the young labor force | 0.7778 | 0.7273 | 1 | 1.1667 | 1.4000 |
| Education level of lower-income households | 1 | 0.5625 | 0.8571 | 1 | 1.88889 |
| Proportion of households receiving NTFPs-related skills training | 1.1250 | 0.5556 | 0.7143 | 0.5294 | 1 |

**Table A9.** C5–D: Judgement matrix of the sustainability of NTFPs industry.

| | The Positive Outlook Rate for NTFPs Industry Development | The Support Rate for the Expansion of the NTFPs Industry | The Support Rate for the Increase in NTFPs Industry Funds |
|---|---|---|---|
| The positive outlook rate for NTFPs industry development | 1.0000 | 1.2857 | 1.2222 |
| The support rate for the expansion of the NTFPs industry | 0.7778 | 1.0000 | 1.0000 |
| The support rate for the increase in NTFPs industry funds | 0.8182 | 1.0000 | 1.0000 |

**Table A10.** C6–D: Judgement matrix of the sustainability of ecological protection.

| | Proportion of Households Using New Energy | The Importance of Ecological Protection |
|---|---|---|
| Proportion of households using new energy | 1.0000 | 1.8889 |
| The importance of ecological protection | 0.5294 | 1.0000 |

**Table A11.** C7–D: Judgement matrix of the satisfaction with poverty alleviation work.

| | Satisfaction with the Poverty Alleviation Policy of the NTFPs Industry | Satisfaction with Poverty Alleviation Projects within the NTFPs Industry | Satisfaction with the Poverty Exit Mechanism |
|---|---|---|---|
| Satisfaction with the poverty alleviation policy of the NTFPs industry | 1.0000 | 1.2857 | 1.2222 |
| Satisfaction with poverty alleviation projects within the NTFPs industry | 0.7778 | 1.0000 | 1.0000 |
| Satisfaction with the poverty exit mechanism | 0.8182 | 1.0000 | 1.0000 |

**Table A12.** C8–D: Judgement matrix of the satisfaction with poverty alleviation effect.

| | Satisfaction with the NTFPs Production Conditions in the Study Area | Satisfaction with the Development of NTFPs Industry | Satisfaction with Ecological Environment Protection |
|---|---|---|---|
| Satisfaction with the NTFPs production conditions in the study area | 1.0000 | 1.2857 | 1.2222 |
| Satisfaction with the development of NTFPs industry | 0.7778 | 1.0000 | 1.0000 |
| Satisfaction with ecological environment protection | 0.8182 | 1.0000 | 1.0000 |

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
