# Peer review of "Has the Development of the Non-Timber Forest Products Industry Achieved Poverty Alleviation? Evidence from Lower-Income Forest Areas in Yunnan Province"

_forests, doi:10.3390/f14040776_

Round 1

Reviewer 1 Report (Previous Reviewer 1)

Lines 25-26: no need '.' before (2) and (3)

Line 42: ...'more and more obvious' - just write 'more obvious'.

Line 82: remove '.' before [21]

Line 157: Although stated as a  sample village (line 165) and described as a typical case, explain why Sanhe Village was chosen as the study site.

Line 168-169: I don't think random sampling is accurately described as a sampling technique in this study.

The way this study was conducted is matching purposive sampling, considering the selection of farmers based on their livelihood and proximity to NTFPs industry in Nujiang.

Line 170: Were there any reasons why only selected 150 households? And from these households, do you interview 1 member of the household or the entire members of 1 household? If doing either 1 of these, explain why?

Line 172-173: These sentences do not accurately reflect the information in Table 2.

Table 2:

'Content' should be called 'variables' while 'Questions' should be called 'scope of variables', or the very least 'Questions on'.

Figure 2: Needs to properly place in the manuscript, it is too close to subsection 2.3.1, and the diagram is too close to the word ' Figure 2..'

Line 307: Remove '.' after [34]

Line 316: Actually, the proportion is not the same.

Ref no 22 need to fix its alignment format

Reviewer 2 Report (Previous Reviewer 2)

the authors have incorporated the minor correction I suggested in my earlier review of this paper. Congratulations to the authors! and wish you continued success with an excellent contribution to society through your research.

Reviewer 3 Report (Previous Reviewer 3)

Dear Authors, 

I appreciate for revising the manuscript. 

I have noted some comments and suggestions on the text and please revise it again. It needs extensive editing in English. Als, Disccuison section need to rewrite carefully. 

All the best,

Reviewer 4 Report (Previous Reviewer 4)

Grammatical flaws.  For example, The survey can be divided into the following states (line 160). Did the authors divide the survey stages or just mentioned they can do those things……….;  Line 182: Future tense; Line 192.

Study Area: Line 123-134: Citation needed

Data Sources: Lines 170: What does the “certain proportion” means? Why authors are not reporting the sample size?

Line 170: The respondents were from poor-class households. Why did the data collector(s) do a survey for poor households only? How many households class were there and why authors choose poor households for the survey? Who did the authors do the questionnaire survey from the households? Was s/he household head or any member of that household? What did the authors do if they did not meet the household head for an interview? Please mentioned clearly.

The paper explained the analysis part in depth but, the survey procedure is not well explained. Approval prior to survey; what type of survey (face-to-face or mail survey or online survey). How many in-depth interviews were taken? What was the purpose of field observation?

Figure 3: Texts are not clear to me. Increase font size.

Line 317: How did 31.8% (51-60 years old) become a majority of the respondents?

Table 9: What there is no age group between 41-50 years old?

Line 333: Typo. Please fix it.

Line 338: The authors mentioned that the final score of 79.33 means the performance of poverty alleviation through NTFPs is good. What was the classification of that score? What is the minimum score to get “good”? or what if it got a 50 score? Was it medium or bad? Please mention it and provide reference.

This manuscript is a resubmission of an earlier submission. The following is a list of the peer review reports and author responses from that submission.

Round 1

Reviewer 1 Report

It is recommended for the manuscript be proofread. 

line 14: write 'p' of 'product' in Capital letter, and add 's' as in 'Products'.

Write 'NTFPs' throughout the manuscripts

Line 28: Check spelling 'sustaonability'

Line 38: Remove one of the '.'

*what is the parameter to test performance for 'ecological environment'?

Line 44: '...

 becoming more and more obvious.' - this sentence lacks significance. Recommend being rephrased by explaining the role of forests in relation to climate change and biodiversity loss.

line 50: check spelling 'policymkers'

Line 73: Is it possible to replace 'forest tourism industry' with 'ecotourism'?

This considering the industry seems to have improved farmers' livelihood while protecting forest resources.

Line 78: delete '.' after 'North America'.

Line 80: Replace 'European' with 'Europe'

Line 80: Why suddenly write about 'Priority Areas'? To be coherent, connect it with the situation in Yunnan Province.

Line 83, Line 132: 'What's more' - replace with a better sentence, avoid the colloquial language

Line 107: spacing need for 'alleviationeffect '

Line 120: Describe the type of survey, when was conducted, and the objective of the survey.

Lines 124-126: Provide statistics to support these findings, and present specific parameters that have been used in the survey that produced these findings.

Lines 135-163: 'get rid of poverty' can be replaced with 'to reduce poverty rate'.

Lines 141-142: check selling 'Aother' , 'huoseholds'.

Section 2.2, from lines 146 - 152: Check the spacing between words.

Line 151: Table 2 should be written as Table 1

Table 1: Replace 'Nation' with 'ethnicity' and add 'level' with 'Education', and add 'School' after 'Middle' and 'High', respectively.

Line 164: Check spelling 'comptant'

Add justifications of why using the judgment matrix?

Table 2 should be improved, as viewers cannot properly see the last column.

Where are the explanations in Table 2?

Line 200: It should be Table 2

Add explanation for results in Table 3

And does n=15 means respondents are 15 or the tested parameters are 15?

Assuming those are parameters, why are there 15 parameters (n) in Table 3, but only 12 in Table 4 - why number of estimated parameters not consistent?

A-B, B1-C etc. - what are these? If using an abbreviation, at least explain in the note below the table the meaning of each abbreviation

Line 209: remove space between 'Province' and '.'.

Line 232: What is CNY? - Write full and put the abbreviation in () when the first time mentioned the CNY.

Reviewer 2 Report

Dear authors

Effective and efficient utilization of forest products in sustainability in an indispensable field of economic activity such as NTFP will be possible with the information obtained with accurate data about these resources. In this respect, this manuscript, the product of your research on a new data processing method that serves this purpose, is invaluable work.

Further, the manuscript is properly written and grammatically sound. Both academic and practical relevance can be found in the problem choice and importance. The literature was appropriately and effectively reviewed. The research's objectives and title are justified by the methods utilised, which are adequate and conventional. Data were methodically gathered in accordance with acceptance criteria. The data was collected in a proper and sufficient manner. Data analysis and interpretation are properly made as per objectives and are appropriate and relevant. The way the result was presented. and discussed with suitable tables and figures with relevant updated literature are good. Measurement, scientific terms properly cited.

Nice work!

Congratulations to the authors! and wish you continued success with a good contribution to society through your research.

 Accept the manuscript for publication after suggested minor corrections!

Point 1

Please write the conclusion after the discussion

Point 2

Scope to improve research gaps and research contribution

Point 3

Scope to improve the value of the study

Point 4

Scope to improve the practical application for the implication

Point 5

Please explain the limitation of the study, and future research direction more clearly.

 winning regards

Reviewer 3 Report

Dear Authors, 

I appreciate your invaluable efforts in writing an invaluable paper by mixing AHP in the evaluation of the role of NTFPs in rural poverty alleviation. 

Although the topic is good I think there is needed extensive consideration for the edition. 

I read it three times but I think it needed a substantial edition on the structure. I think most of the methods are in the results and vice versa. I wrote some comments and suggestions for improving the manuscript.

Please revise all of them carefully.

All the best

Reviewer 4 Report

Thank you, authors and editor, for providing an opportunity to read this manuscript. The authors attempted to write the manuscript. The manuscript has valuable data and information that can be useful for forest managers, policymakers, farmers, and industries. However, there is a presence of serious flaws. Especially, a research design is not satisfactory. I encourage authors to rewrite or revise and resubmit the manuscript in the journal.

Major Comments:

1.      Extensive grammatical errors and typos.

2.      The paper does not follow the standard format of a scientific paper. Here, I see the Result and Analysis are combined in one section, and the Discussion section is after the Conclusion section.

3.      Lack of appropriate survey design.

4.      Study area: The description of the study area is more generic. I suggest including specific characteristics of it. For example, location, climatic condition, socio-demographic features, people’s dependency on forest resources (NTFPs), economic status (no. of NTFPs industry, govt. investment amount, etc.), and ecological or environmental issues of the area.

5.      Data sources: It is not well-explained. Some results are included in the data source. Please separate results from the data source. I suggest rewriting the data source clearly and explicitly. Describe the survey instrument.

6.      Flawed methodology. Survey approach. How did the authors select respondents (just saying relevant personnel is not sufficient)? List any criteria, if followed. The authors mention two Evaluation Index Systems. However, there is no background of the systems (e.g., who discovered or designed the system? 

7.       The data analysis section is combined with the results section, which is uncommon in a scientific paper.

Specific comments:

Lines 101-103: The forest area of its reached 20.2……….. 65.4%. Is it increased or decreased?

Line  118: What is the area of Sanhe village (study area)? Since the study was about poverty alleviation, it is important to provide socio-demographic information about the study area.

Figure 1: Title of the figure. Is it county or country?

Lines 141-142: Presence of typo

Table 8: How many respondents were there in question? Does each question have the same sample size?